# Nondestructive detection of photonic qubits

Dominik Niemietz[1 ✉], Pau Farrera[1,2], Stefan Langenfeld[1] & Gerhard Rempe[1]

One of the biggest challenges in experimental quantum information is to sustain the fragile superposition state of a qubit[1]. Long lifetimes can be achieved for material qubit carriers as memories[2], at least in principle, but not for propagating photons that are rapidly lost by absorption, diffraction or scattering[3]. The loss problem can be mitigated with a nondestructive photonic qubit detector that heralds the photon without destroying the encoded qubit. Such a detector is envisioned to facilitate protocols in which distributed tasks depend on the successful dissemination of photonic qubits[4,5], improve loss-sensitive qubit measurements[6,7] and enable certain quantum key distribution attacks[8]. Here we demonstrate such a detector based on a single atom in two crossed fibre-based optical resonators, one for qubit-insensitive atom–photon coupling and the other for atomic-state detection[9]. We achieve a nondestructive detection efficiency upon qubit survival of 79 ± 3 per cent and a photon survival probability of 31 ± 1 per cent, and we preserve the qubit information with a fidelity of 96.2 ± 0.3 per cent. To illustrate the potential of our detector, we show that it can, with the current parameters, improve the rate and fidelity of long-distance entanglement and quantum state distribution compared to previous methods, provide resource optimization via qubit amplification and enable detection-loophole-free Bell tests.

The qubit is the elementary information unit in quantum information science[1]. Encoding a qubit into two modes of a single photon allows the distribution of quantum information over long distances. This has enabled a range of experiments, from fundamental tests of quantum physics[10,11] to applications related to quantum communication[3] and quantum networks[9,12]. However, unavoidable absorption, diffraction and scattering losses of long transmission channels severely limit the transfer distance. It is important to emphasize that these losses typically occur independent of the state of the qubit that is encoded in the two optical modes, be it time bins or light polarization. In fact, in optical fibres the loss rate of the qubit carrier, the photon, can be many orders of magnitude larger than all decoherence rates of the encoded qubit. By contrast, material qubit carriers are rarely lost in any quantum information protocol demonstrated so far. For photons, the loss is fundamental and cannot be eliminated in any of the envisioned quantum information processing tasks. However, the loss effect could be mitigated by tracking the photon without destroying the encoded qubit, provided a nondestructive photonic qubit detector (NPQD) is available.

Once at hand, such an NPQD could communicate to both the sending and the receiving nodes of a quantum network whether or not a photonic qubit has been lost along the way (Fig. 1a). This loss monitoring has several advantages: first, it allows quantum communication schemes to stop the execution of further operations and restart the protocol when the qubit-carrying photon was found to be lost along the communication channel. This is important when these operations are time-expensive or involve the use of precious resources such as long-distance entanglement for quantum teleportation. Second, the detector can precertify the arrival of a photonic qubit immediately before a planned quantum measurement so that the latter is only conducted when the qubit is present at the measurement input. This enables us to perform loss-sensitive qubit measurements even in the presence of high transmission losses. Examples are measurements limited by the detection noise, photon-receiving repeater schemes[13], or experiments where qubit loss leads to loopholes as in Bell tests[6,7]. The last are of key importance for device-independent quantum key distribution[14].

Despite recent progress on the quantum nondemolition detection of optical and microwave fields[15–21], the nondestructive detection of the photonic energy without projection of the encoded qubit information remains an outstanding challenge. Some work related to the experimental implementation of NPQDs in the optical domain has been conducted, such as the nondestructive detection of bright two-mode light pulses using cross-phase modulation[22], although this scheme operates far from the single-photon qubit regime. Another approach makes use of parametric down-conversion of the incoming qubit photon[23], where one down-converted photon heralds the qubit presence and the other provides the qubit information. This approach shows a severely limited efficiency (−76 dB), owing to the small conversion efficiency of the applied photon-splitting process. Other related work includes heralded qubit amplifiers, which use two ancilla photons that interfere with the input signal[24]. However, interference requires prior knowledge about the pulse shape and arrival time and therefore

[1]Max-Planck-Institut für Quantenoptik, Garching, Germany. [2]ICFO—Institut de Ciencies Fotoniques, The Barcelona Institute of Science and Technology, Castelldefels, Spain. ✉e-mail: dominik.niemietz@mpq.mpg.de

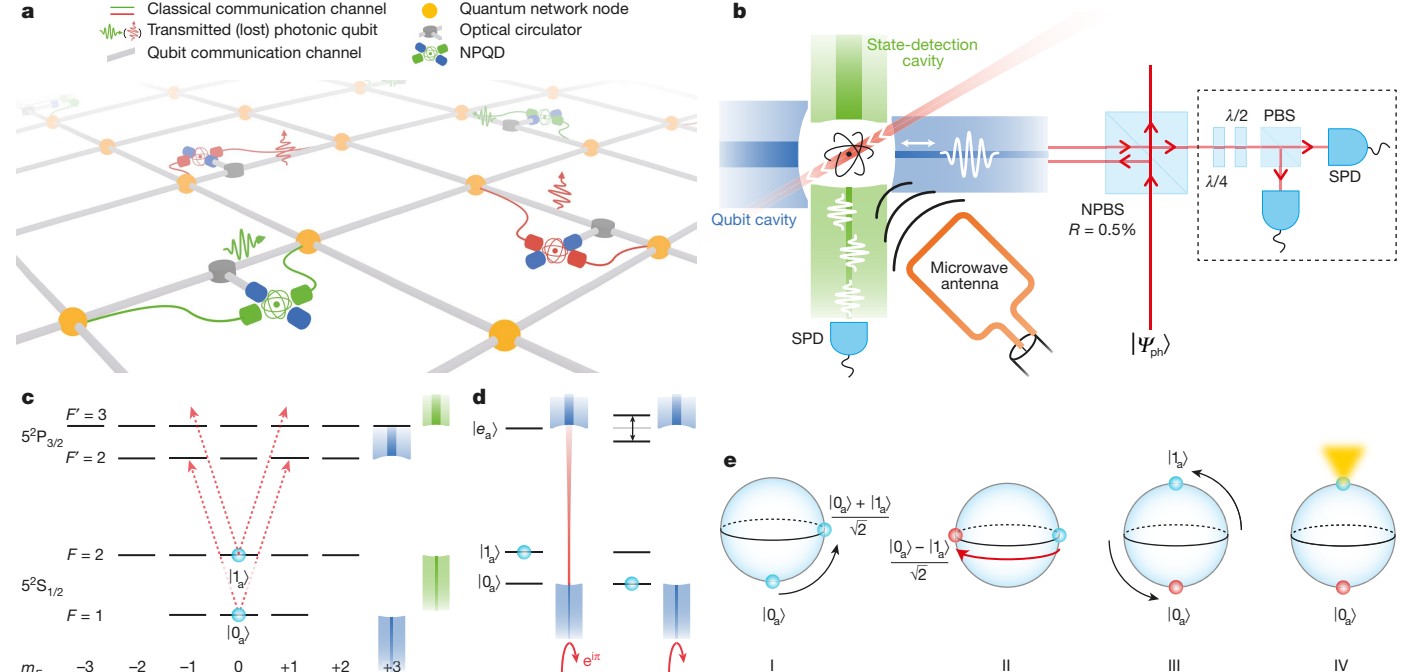

**Fig. 1 | Nondestructive photonic qubit detector. a**, A quantum network including NPQDs. Detected and non-detected qubit photon events are coloured in green and red, respectively. **b**, The NPQD set-up showing the crossed optical-fibre cavities together with the microwave antenna and the state-detection beam (red solid line). Input photonic qubits (red arrows) are reflected by the NPQD via a highly transmittive non-polarizing beam splitter (NPBS, with a reflection of $R = 0.5\%$) and later sent to a qubit measurement set-up (black dashed rectangle) including $\lambda/2$ and $\lambda/4$ waveplates, a polarizing beam splitter (PBS) and single-photon detectors (SPD). **c**, Atomic-level scheme together with the qubit coupling transitions (dashed arrows). **d**, The atomic state affects the entrance of the qubit photon into the qubit cavity. $|e_a\rangle$ denotes the atomic excited state that is coupling to $|0_a\rangle$ via the qubit cavity modes. **e**, Bloch spheres I–IV show the atomic state for different stages of the ideal NPQD scheme.

restricts the range of possible applications. Finally, our previous work on heralded quantum memories[25,26] can, in combination with readout, be used to herald photonic qubits. But the single-photon herald signal is highly sensitive to losses and the system therefore has a small detection efficiency. It is important to point out that (in contrast to the NPQD reported here) all mentioned qubit-heralding schemes destroy the original incoming photon waveform, which is important in applications where synchronization and interference matter, as for time-bin qubits or in linear optical quantum computing[27].

We here demonstrate the implementation of an NPQD using a single atom strongly coupled to the modes of two independent optical resonators. Photonic polarization qubits are sent onto the qubit cavity where they are reflected while imprinting a $\pi$ phase shift on an atomic superposition state. By coherent manipulation of the atomic superposition state, phase information is mapped into population information, which can be read out using the state-detection cavity. This signal heralds the nondestructive detection of the photonic qubit reflected off the qubit cavity. We show that the strong atom–photon interaction provided by both cavities allows for an efficient NPQD. We also prove that the reflection mechanism preserves the temporal waveform of the light pulse, which is of key importance for photonic time-bin qubits.

The qubit–atom interaction mechanism is, in essence, a single-rail version of the scheme proposed by Duan et al.[28]. In our case, it starts with a $^{87}$Rb atom prepared in state $(|0_a\rangle + |1_a\rangle)/\sqrt{2}$ (Bloch sphere I in Fig. 1e). The atomic state $|0_a\rangle$ is strongly coupled to the optical cavity modes, thus preventing a photon from entering the cavity. Instead, the photon reflects from the cavity input mirror (Fig. 1d) so that $|\Psi_{ph}\rangle|0_a\rangle \rightarrow +|\Psi_{ph}\rangle|0_a\rangle$, where $|\Psi_{ph}\rangle$ denotes a photonic polarization qubit. By contrast, state $|1_a\rangle$ has no transition in resonance with the qubit cavity. Hence, the qubit photon enters the cavity and acquires a $\pi$ phase shift upon reflection, $|\Psi_{ph}\rangle|1_a\rangle \rightarrow -|\Psi_{ph}\rangle|1_a\rangle$. Together, the qubit photon flips the atomic superposition state to $(|0_a\rangle - |1_a\rangle)/\sqrt{2}$ (Bloch sphere II).

Finally, a $\pi/2$ pulse rotates the atomic state to $|0_a\rangle$ ($|1_a\rangle$) in the case of one (no) qubit photon (Bloch sphere III), enabling us to witness the atomic phase flip by means of deterministic atomic-state detection (Bloch sphere IV).

Compared to previous experiments[18], our scheme is made possible by two key ingredients: first, we choose a state $|0_a\rangle := |F = 1, m_F = 0\rangle$ so that the qubit cavity, with its two left and right circularly polarized and frequency-degenerate eigenmodes, couples strongly to two atomic transitions, $|0_a\rangle \leftrightarrow |F' = 2, m_F = \pm 1\rangle$ (Fig. 1c). Here, $F$ and $F'$ denote the magnitude of the total atomic angular momentum and $m_F$ denote their projection onto the quantization axis. The coupling strengths for both transitions are equal by definition but much smaller than for a cycling transition. We compensate this reduction by using a miniaturized fibre resonator[29], achieving a coupling constant $g = 2\pi \times (18.6 \pm 0.5)$ MHz and a cooperativity of $C = 1.67 \pm 0.09$ (see Methods). Second, the compact lateral size of fibre resonators allows for the integration of a second crossed optical-fibre resonator with small mode volume. Here, it is used as the state-detection cavity which is tuned to the atomic cycling transition $|F = 2\rangle \leftrightarrow |F' = 3\rangle$, enabling a strong fluorescence signal to be observed from $|1_a\rangle := |F = 2, m_F = 0\rangle$. This is used to distinguish the two atomic states $|0_a\rangle$ and $|1_a\rangle$ with a fidelity of $(98.2 \pm 0.2)\%$ at a photon-number threshold of one.

The physical elements needed to implement the NPQD are shown in Fig. 1b. The photon qubit $|\Psi_{ph}\rangle$ first passes a highly transmitting non-polarizing beam splitter. Although the non-polarizing beam splitter is suitable for characterization measurements employing weak coherent pulses, for future implementations using single photons it should be replaced by an efficient optical circulator (Fig. 1a). After reflection at the non-polarizing beam splitter, the photon qubit is coupled into a single-mode optical fibre that connects to the qubit fibre cavity where the nondestructive interaction takes place. After reflection, the qubit photon is characterized by state tomography.

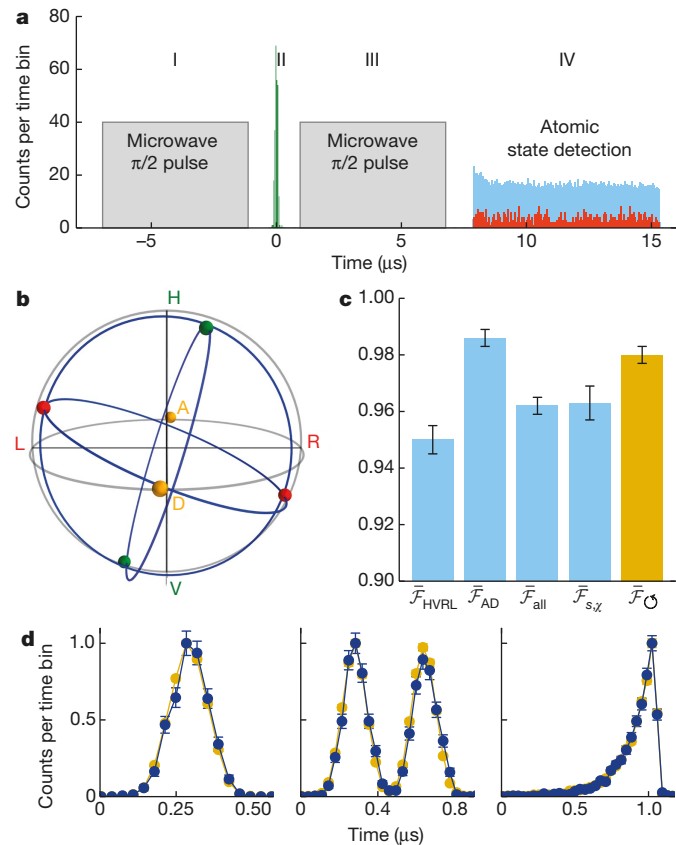

**Fig. 2 | Experimental NPQD implementation. a**, Time histogram showing the NPQD sequence, which includes the microwave pulses, the reflected qubit photon events (green) and the state-detection photon events. Red (blue) coloured state-detection counts represent data conditioned on a reflected photonic qubit detection (without an incoming photonic qubit). Labels I–IV refer to the Bloch spheres in Fig. 1e. **b**, Poincaré sphere showing the result of state tomography (coloured spheres) for reflected photonic qubits that were nondestructively detected. Labels indicate input polarizations. **c**, Bar chart of mean fidelities $\overline{\mathcal{F}}_x$ and the quantum process state fidelity $\overline{\mathcal{F}}_{s,\chi}$. The Poincaré sphere shows a rotation around the $A/D$ axis which, if eliminated, leads to $\overline{\mathcal{F}}_{\circlearrowright}$. **d**, The preservation of the photonic waveform is tested by reflecting photonic qubits with three different waveforms off the NPQD (blue points). Yellow points represent empty cavity measurements as reference. The error bars in **c** (**d**) describe the $1\sigma$ confidence interval (standard deviation). The error bar of $\overline{\mathcal{F}}_{s,\chi}$ is assessed with a Monte Carlo method and represents the standard error.

Coherent manipulation of the atomic ground states is achieved by means of microwave fields emitted by an antenna close to the fibre cavities. The microwave flips the atomic state in about 12 μs with a probability of $(97 \pm 1)\%$ (see Methods). The detector herald signal is generated by a laser that, together with the state-detection cavity, induces cavity-enhanced fluorescence[9] for typically 7.5 μs. The fluorescence photons are predominantly emitted into the cavity from where they are guided towards a single-photon detector. Note that both fibre cavities are single-sided (for details, see ref. [26]), so that the dominant escape channels for cavity photons are the two single-mode fibres.

We now characterize the nondestructive detection of photonic polarization qubits. For this, weak coherent pulses at the single-photon level were used. Figure 2a illustrates the NPQD temporal sequence by showing a time histogram of single-photon detector counts at the qubit measurement setup and the state-detection cavity output. The mean photon number here is $|\alpha|^2 = 0.13$ in front of the qubit cavity. As can be seen, the reflection of photonic qubits at the NPQD (green bars) has a large effect on the number of observed state-detection photons (red

bars) compared to the situation where no qubits are sent (blue bars). This enables an efficient detection of the photonic qubits. However, the definition of the NPQD efficiency is ambiguous as it depends on the use case. Relevant parameters that relate to the NPQD efficiency include the conditional probability of detecting the atom in the qubit-heralding state $|0_a\rangle$, either given a qubit at the qubit cavity output $(1_{oq})$, $P(0_a|1_{oq}) = (79 \pm 3)\%$, or given a qubit at the NPQD input $(1_{iq})$, $P(0_a|1_{iq}) = (45 \pm 2)\%$, and the photon survival probability $\eta_{surv} = (31 \pm 1)\%$. Another figure of merit for any detector is the dark-count probability, which is $p_{DC} = (3.3 \pm 0.2)\%$ for our NPQD.

As NPQDs should preserve the qubit, polarization state tomography was conducted on the outgoing photonic qubits and compared with the incoming qubits. With this we calculate the fidelity $\mathcal{F} = \langle \Psi_{in}|\rho_{out}|\Psi_{in}\rangle$. Six orthogonal polarization states were tomographically investigated (Fig. 2b), showing an overall mean state fidelity of $\overline{\mathcal{F}}_{all} = (96.2 \pm 0.3)\%$ conditioned on a nondestructive qubit detection. Our fidelity, here for $|\alpha|^2 = 0.2$, greatly exceeds the upper limit of the fidelity 2/3 or 5/6 which is possible for schemes employing $1 \to \infty$ or $1 \to 2$ universal quantum cloning machines, respectively[30]. Note that the limits must be slightly shifted to 67.5% (ref. [31]) and 84.9%, owing to higher photon number contributions of weak coherent pulses. Against this backdrop we conclude that our NPQD operates in the quantum regime.

A major contribution to the infidelity comes from a polarization rotation around the $A/D$ axis, attributed to a residual birefringence of the qubit cavity (see Methods). The mean fidelity for rotated and non-rotated states is given by $\overline{\mathcal{F}}_{HVRL}$ and $\overline{\mathcal{F}}_{AD}$, respectively (Fig. 2c). However, taking into account this rotation in the calculation (which would be experimentally feasible by placing retardation waveplates after the qubit cavity) leads to an overall mean state fidelity of $\overline{\mathcal{F}}_{\circlearrowright} = (98.0 \pm 0.3)\%$ (orange bar). The remaining 2% infidelity is attributed to errors in the calibration of the qubit measurement setup and to fluctuations of the qubit cavity resonance frequency. Additionally, a quantum process is reconstructed via a maximum-likelihood fit, leading to an overall mean state fidelity of $\overline{\mathcal{F}}_{s,\chi} = (96.3 \pm 0.6)\%$.

Another important property of an NPQD is to preserve the temporal waveform of the photonic qubit (Fig. 2d). This is of great importance in scenarios where photons interfere or where quantum information is encoded in the temporal mode of the photon in addition to, or instead of, the polarization. To illustrate that our NPQD preserves the waveform, we compare three different envelopes of qubit pulses reflected off our atom–cavity system (blue points) with pulses reflected off an empty cavity (yellow points). For all three cases we find an intensity waveform overlap between the outgoing and the incoming photon exceeding 99.5%.

The mean input photon number $|\alpha|^2$ of the weak coherent pulses encoding the qubit has a role in the performance of the NPQD. Its characterization is presented in Fig. 3 where different figures of merit of our detector are shown. For the given range of $|\alpha|^2$ in front of the qubit cavity, the nondestructive detection probability upon qubit survival $P(0_a \geq 1_{oq})$ (blue points in Fig. 3a) shows a maximum of $(79 \pm 3)\%$ at $|\alpha|^2 = 0.13$. Here, the major error is attributed to a difference in intensity reflection coefficients for a coupling $(R_{|0_a\rangle} = 0.50 \pm 0.02)$ and a noncoupling atom $(R_{|1_a\rangle} = 0.117 \pm 0.003)$; see Methods. For high $|\alpha|^2$, the conditional probability converges towards 0.5 owing to the balanced contribution of odd and even photon numbers[32], and decreases for small $|\alpha|^2$ values owing to dark counts of the qubit measurement set-up. The unconditional probability $P(0_a)$ (green points) shows the same convergence for high $|\alpha|^2$. However, in the limit of $|\alpha|^2 \to 0$, the probability is lower-bounded by $p_{DC}$ (grey dashed line), which is due to imperfections in the atomic-state manipulation, in the optical pumping and in the state detection.

Another important figure of merit is the probability of having a photonic qubit at the NPQD output conditioned on its nondestructive detection, $P(1_{oq}|0_a)$. Because we characterize the NPQD with weak coherent states (and not with single-photon Fock states), we instead show

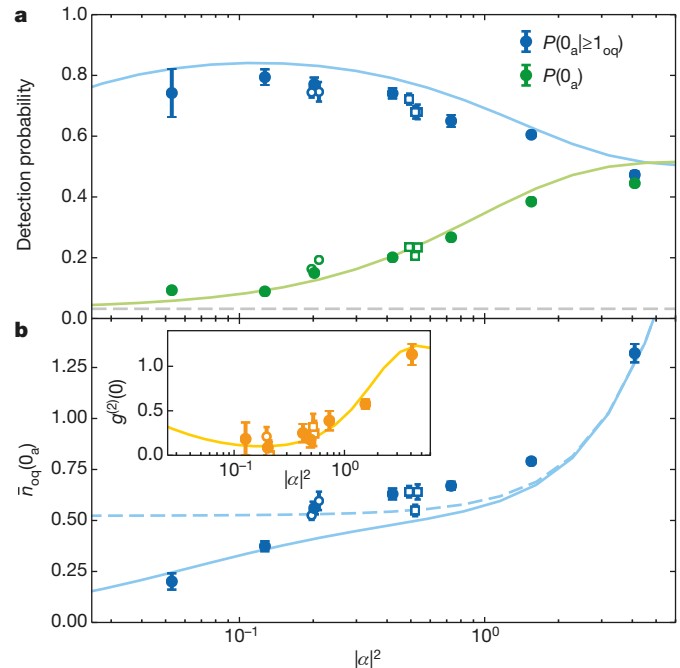

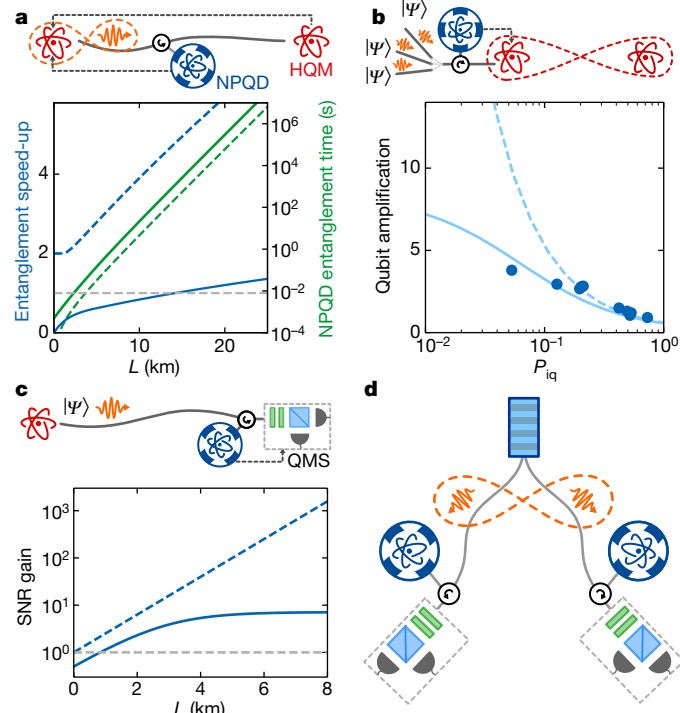

**Fig. 3 | Characterization with different mean input photon numbers.**
**a**, Nondestructive detection probability as a function of the mean input photon number $|\alpha|^2$ in front of the qubit cavity. $P(0_a|\geq1_{oq})$ is conditioned on a successful qubit reflection ($\geq1_{oq}$), whereas $P(0_a)$ is unconditioned. The horizontal dashed line represents the NPQD dark-count probability $p_{DC}$.
**b**, Mean photon number at the qubit cavity output conditioned on a nondestructive detection event $\bar{n}_{oq}(0_a)$. Inset, the autocorrelation function $g^{(2)}(\tau=0)$ of the nondestructively detected qubits. All solid data points are taken with Gaussian photon pulse shapes and linear near-vertical polarization. Open circle (square) data points were taken with orthogonal polarization (different pulse shapes; Fig. 2c). For all plots the solid line is given by theoretical simulations described in Supplementary Information. The dashed line in **b** considers $p_{DC}=0$. The error bars in **a** represent the $1\sigma$ confidence interval. The error bars in **b** and the inset to **b** represent the standard error and the standard deviation, respectively.

**Fig. 4 | NPQD applications. a**, Entanglement generation between an atom–photon entanglement source and a heralded quantum memory (HQM). An NPQD along the transmission channel can speed up the mean sender–receiver entanglement time compared to the standard situation. The blue and green coloured lines represent the entanglement speed-up and the NPQD entanglement time, respectively. **b**, Photonic qubits are remotely sent to a receiver for a follow-up operation that involves precious resources (for example, long-distance entanglement). The qubit amplification gives the ratio of the probability of having a photon after its nondestructive detection at the NPQD to the probability of having a photon before detection ($P_{iq}$). **c**, Photonic qubits transfer to a noisy qubit measurement set-up (QMS). Gating the QMS with the NPQD herald signal improves the SNR. **d**, Bell test based on precertification of the photon's presence[6]. The dashed lines in the plots represent the situation of a perfect NPQD (**a**, **c**) or a dark-count-free NPQD (**b**).

in Fig. 3b the conditioned mean photon number, $\bar{n}_{oq}(0_a)$, which is equivalent to $P(1_{oq}|0_a)$ for small input photon numbers. We observe $\bar{n}_{oq}(0_a)=0.56\pm0.02$ for $|\alpha|^2=0.2$, but this value decreases for smaller $|\alpha|^2$, owing to the NPQD dark counts $p_{DC}$. The simulation that considers $p_{DC}=0$ (dashed line), converges towards $P(1_{oq}|0_a)=52.3\%$ for $|\alpha|^2\to0$. This value is below one because of parasitic losses of the cavity mirrors, imperfect mode matching and a finite atomic decay rate. Interestingly, not only the mean photon number but also the photon statistics change after qubit reflection (Fig. 3b, inset). We prove this by measuring the second-order autocorrelation function at zero time delay, $g^{(2)}(0)$, of the reflected qubits conditioned on their nondestructive detection. The obtained sub-Poissonian statistics $g^{(2)}(0)<1$ originates from the distillation of single photons out of the incoming weak coherent pulse (see Supplementary Information and ref. [32]).

To explore the potential of our NPQD, we discuss now four exemplary applications that would benefit from our detector (details in Methods). Figure 4a, b illustrates situations in which monitoring the qubit loss along a transmission channel saves time and precious resources. The first example (Fig. 4a) consists of an atom–photon entanglement source[9] that sends photonic qubits to a heralded quantum memory[26] to generate sender–receiver entanglement. The plot shows the entanglement speed-up defined as the ratio of the mean entanglement generation time without the nondestructive detector to that with the nondestructive detector, $T_{ent}/T_{ent}^{NPQD}$, versus the channel length $L$. The position of the detector was chosen such that $T_{ent}^{NPQD}$ is minimal.

Our nondestructive detector (solid lines) outperforms direct transmission at channel distances greater than or equal to 14 km, whereas a perfect NPQD (dashed lines) would provide an advantage at any distance. The second example (Fig. 4b) describes a situation in which it is more efficient to perform a complex operation on an input photonic qubit only if this qubit has survived the transmission through a lossy channel. This can be accomplished by placing an NPQD right before the operating node. One example for a complex operation is quantum teleportation of the photonic qubit by means of previously established long-distance entanglement. A good figure of merit is the ratio of the probability of having a reflected qubit photon after nondestructive detection $P(1_{oq}|0_a)$ to the probability of having an incoming qubit photon $P_{iq}$, usually called qubit amplification[24]. The qubit amplification is substantially higher than one for $P_{iq}\ll1$ and would notably improve in the absence of NPQD dark counts (dashed line).

Another group of applications refers to situations in which NPQDs can improve a subsequent photonic qubit measurement. Figure 4c shows a scenario in which photonic qubits are sent to a remote receiver that uses noisy detectors for the qubit measurement. An NPQD right before the receiver enables measurement of the qubit only when it is not lost, reducing the impact of measurement noise. This is of particular interest for quantum key distribution, because the dark-count noise of classical detectors is an important limitation on the quantum key rate for large distribution distances[14]. The signal-to-noise ratio (SNR)

gain is defined as the ratio of the SNR with an NPQD to that without an NPQD, $SNR_{NPQD}/SNR$, and exceeds one for a sender–receiver distance $L > 1$ km. For longer distances, this ratio converges towards approximately seven, which could be raised with lower NPQD dark counts (illustrated by the dashed line, which considers perfect NPQD parameters). A last example relates to loophole-free Bell tests (Fig. 4d). As proposed previously[6], the herald signals of NPQDs could allow two parties to be certain that they share an entangled photon pair right before the measurement, helping to close the detection loophole. The important parameter here is the photonic qubit reflection probability conditioned on its nondestructive detection. From our experiment we find $\bar{n}_{oq}(0_a) = 0.56 \pm 0.02$ for an input $|\alpha|^2 = 0.2$ (Fig. 3b). This value exceeds the minimum detection efficiency of 43% (assuming no detector background noise) which is required for a detection-loophole-free asymmetric Bell test[33].

In conclusion, we have demonstrated a nondestructive detector for photonic polarization qubits. We anticipate that it should also work with time-bin qubits. The main features are a conditional detection efficiency of up to $(79 \pm 3)\%$ and a qubit preservation fidelity of at least $(96.2 \pm 0.3)\%$. None of the observed limitations seems fundamental. Most importantly, we present four possible applications that would benefit from our present device, to which we still expect to make improvements. This creates confidence that our detector will turn out to be a useful tool in near-future quantum communication links and fundamental tests of quantum physics.

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

# Methods

## NPQD experimental sequence

The experiment starts with a two-second-long loading phase of a single atom into the crossing point of the two fibre-cavity modes (details in ref. [26]). A magneto-optical trap with $^{87}$Rb atoms is loaded approximately 10 mm above the fibres and is released such that the laser-cooled atoms fall towards the intracavity region. At the crossing point of the two cavity modes there is a three-dimensional optical lattice where single atoms are trapped. The lattice consists of two blue-detuned intracavity traps (774.6 nm and 776.5 nm wavelength) and one red-detuned standing-wave optical dipole trap (799.2 nm wavelength), detuned with respect to the $D$ line and with a trap depth of $U_0/k_B \approx 1$ mK each ($k_B$, Boltzmann constant). A cooling beam comes in at an oblique angle and cools the single atoms in the trap, where they remain for a few seconds. Cooling light that is scattered by the atom into the state-detection cavity mode is subsequently detected with a single-photon detector so that the presence of single atoms is confirmed. Afterwards, the NPQD scheme runs with a repetition rate of 576 Hz. This sequence is divided into three parts: the atomic cooling, atomic-state preparation and the NPQD part. The temporal length of atomic cooling constitutes 97% of the fast sequence time. This high fraction goes along with a desired and small microwave duty cycle to minimize system heating and cavity shaking that is attributed to the applied microwave fields. However, during this long cooling time, we conduct six 3-µs-long sequences to generate single photons emitted by the atom. The recorded photon counts are used to calculate the autocorrelation function $g^{(2)}(0) = 1 - 1/n$, to infer the number of trapped atoms $n$. The atom cooling is followed by a 30-µs-long optical pumping phase to prepare the atom in the Zeeman state $|F=1, m_F=0\rangle$. Here, the applied laser fields are the same as described in ref. [26], except we do not apply the final 4-µs-long π-polarized laser field used in that work. After optical pumping, the sequence starts with the NPQD scheme as reported in the main text. We have stated that for cavity-assisted state detection the state-detection cavity is near-resonant with the atomic transition $D_2$: $|F=2\rangle \leftrightarrow |F'=3\rangle$. Because the state-detection cavity exhibits an intentional polarization mode splitting by more than four cavity linewidths [26], only the π-polarization mode is near-resonant with the atomic transition; the second polarization mode is blue-detuned.

## Coherent atomic-state manipulation

The nondestructive detector scheme requires the coherent manipulation of the atomic ground states $|F=1, m_F=0\rangle$ and $|F=2, m_F=0\rangle$. In our experiment this manipulation is done via microwave radiation. It is advantageous to use microwave fields as opposed to a Raman transition with near-infrared radiation fields, as the latter is intrinsically less detuned from the excited states and therefore leads to a higher probability of populating these, which causes decoherence processes [34].

A microwave antenna is placed inside the vacuum chamber a distance of <2 cm from the crossing point of the fibre-cavity modes. The antenna consists of a single loop with a circumference of 4.4 cm, equal to the microwave field wavelength. A magnetic guiding field of 226.5 mG magnitude defines the quantization axis along the qubit cavity axis which enables us to address a single ground state transition at a microwave pulse duration of 5.8 µs. Extended Data Fig. 1a shows a microwave spectrum for an atom that is prepared in the state $|F=1, m_F=0\rangle$ and which was subsequently driven with a microwave field of varying frequency and a rectangular pulse duration of 25 µs. Final cavity-assisted state detection measures the population in state $5^2S_{1/2} F=2$.

Choosing the related frequency enables the driving of the transition of interest: $|F=1, m_F=0\rangle \leftrightarrow |F=2, m_F=0\rangle$. The coherent Rabi flopping of these two states is shown in Extended Data Fig. 1b, revealing the π/2-pulse duration of 5.8 µs. One limitation that affects this coherent driving is decoherence, which leads to a shrinking of the visibility of the Rabi oscillations. To further characterize this effect, we performed a Ramsey-type experiment (Extended Data Fig. 1c). We have applied two rectangularly shaped π/2 pulses with varying time gap in between. During the waiting time, the internal microwave clock is shifted by 100 kHz to distinguish the decoherence from small microwave detunings. Decoherence is then unambiguously detected by the shrinking visibility of the oscillations. Extended Data Fig. 1c shows how the visibility decreases to the 1/e value (e, Euler's number) after 121 µs, which we define as the coherence time. We suspect that the decoherence arises from the effects of a mechanical non-ground state cooled atom that is trapped via a near-resonant red-detuned optical dipole trap. Hence, the hyperfine ground states $|F=1\rangle$ and $|F=2\rangle$ are differently a.c. Stark-shifted which leads to different trap frequencies and ultimately to non-degenerate ground state transition frequencies. This situation can be improved by ground-state cooling of the single atom [35] or a further red-detuned (and therefore more powerful) optical dipole trap.

## Atom–cavity interaction and conditional reflection

One enabling ingredient for the NPQD is the polarization-independent strong coupling between the atom and the photonic qubit mediated by the qubit cavity. For characterization, reflection spectra were measured (Extended Data Fig. 2a) with a probe field in a superposition of left and right circular polarization, once with an atom coupled to the cavity mode (blue points) and once without any atom (green points). In the figure, a probe field frequency of zero corresponds to the atomic transition frequency. The empty cavity spectrum yields a field decay rate of $\kappa_{QC} = 2\pi \times (34.6 \pm 0.3)$ MHz at a qubit cavity length of 162 µm. The normal-mode spectrum provides a coupling rate of $g = 2\pi \times (18.6 \pm 0.5)$ MHz which leads to—considering the atomic dipole decay rate of $\gamma = 2\pi \times 3$ MHz—a cooperativity of $C = g^2/(2\kappa_{QC}\gamma) = 1.67 \pm 0.09$. Note the difference between the intensity reflection coefficients with ($R_{|0_a\rangle} = 0.50 \pm 0.02$) and without ($R_{|1_a\rangle} = 0.117 \pm 0.003$) a coupling atom, both close to zero detuning. The intensity reflection coefficients at zero atom and cavity detuning can be calculated according to $R = \left| 1 - \mu_{fc}^2 \frac{2\kappa_{QC,1}}{\kappa_{QC}} \frac{1}{2C+1} \right|^2$, which employs the cavity field decay rate via the outcoupling mirror $\kappa_{QC,1} = \kappa_{QC} \times \frac{340 \text{ ppm}}{430 \text{ ppm}}$. The equation for the intensity reflection coefficient additionally considers the mode matching between the fibre and the cavity mode, $\mu_{fc} = 0.92e^{-i0.03}$, with respect to the equation provided in ref. [9]. The difference between the coefficients constitutes a major error source of our nondestructive detector, which leads to a lower detection probability, as explained in the following.

The theoretical model given in Supplementary Information includes a detailed set of contributing imperfections of our NPQD. However, here we assume a perfect system detecting single photons with only the conditional reflection as imperfection to clarify its consequences for the detector. The NPQD scheme starts with the atom prepared in a superposition of two ground states $|\Psi_{a1}\rangle = (|0_a\rangle + |1_a\rangle)/\sqrt{2}$ which ideally turns into $|\Psi_{a2}\rangle = (|0_a\rangle - |1_a\rangle)/\sqrt{2}$ upon successful photon reflection (blue and first red vector in Extended Data Fig. 2b). However, owing to the different cavity reflection coefficients corresponding to the two atomic states, the initial state turns into

$$|\Psi_{a2}\rangle = \frac{r_0|0_a\rangle + r_1|1_a\rangle}{\sqrt{|r_0|^2 + |r_1|^2}} \tag{1}$$

with $r_0 = |\sqrt{R_{|0_a\rangle}}|$ and $r_1 = |\sqrt{R_{|1_a\rangle}}|e^{i\pi}$ (first green vector). As $r_0$ and $r_1$ are not only different in phase but also in magnitude, the atomic state leaves the equatorial plane of the Bloch sphere. A subsequent microwave π/2 pulse rotates this state into $|\Psi_{a3}\rangle = \hat{R}_a(\pi/2)|\Psi_{a2}\rangle$ which goes beyond the pole $|0_a\rangle$ (second green vector) and therefore leads to a state that is not orthogonal to $|1_a\rangle$. The detection probability upon qubit reflection is then upper-bounded by $P(0_a|1_{oq}) = |\langle 0_a|\Psi_{a3}\rangle|^2 = 89\%$. The further reduction of this value that we observe is attributed to a group of imperfections that are further discussed in Supplementary Information.

## Photonic qubit fidelity and cavity birefringence

As shown in Fig. 2, the photonic polarization qubit experiences a small rotation after its nondestructive detection. This is due to the birefringence of the qubit cavity, which originates from a small ellipticity of the fibre-cavity mirrors and leads to a polarization mode splitting[36] of a fifth of the qubit cavity linewidth. By means of a $\lambda/2$ retardation waveplate positioned after the cavity, all polarization states rotate such that the qubit cavity eigenmode polarizations at reflection rotate into polarizations $A$ and $D$ at the detection setup. We measure that a superposition of polarizations $A$ and $D$ experiences a rotation of 42° around the $A/D$ axis on the Bloch sphere in the case of an empty cavity. The detector scheme relies on an atom being in a superposition of a coupling and noncoupling state, and so the incoming photonic qubit does not experience this full rotation. A rotation angle of 19.6° is extracted from a fit of the input and output polarization states which are measured by polarization state tomography (Extended Data Fig. 3a; the analysis follows the description of ref. [26]). This rotation is also observed in Extended Data Fig. 3b, which shows the matrix of the related underlying quantum processes of the detector[37]. The shown matrix is derived from a maximum-likelihood fit with matrix element uncertainties that are calculated according to ref. [26]. Major contributing matrix elements are $\chi_{0,0}$, $\chi_{1,0}$ and $\chi_{0,1}$, illustrating the polarization rotation effect.

In this Article we provide the mean state fidelities under the condition of nondestructive detection. However, evaluation of unconditional SPD counts (for example, events from all photonic qubits reflected from the NPQD) reveals a reduction of the polarization state fidelity for cavity noneigenstate polarizations (compare $\mathcal{F}_{\text{cond.}}$ with $\mathcal{F}_{\text{uncond.}}$ in Extended Data Table 1). We attribute this to a partial entanglement effect between the photonic polarization state and the atomic state, because the polarization rotation preferably occurs when the atom is in the noncoupling state. This effect can be theoretically described as follows. An initial atomic superposition state and an incoming photonic qubit lead to a partially entangled state after reflection,

$$|\psi\rangle = \frac{r_0|\Psi_{\text{ph}}\rangle|0_a\rangle + r_1(\hat{R} \otimes \hat{1})|\Psi_{\text{ph}}\rangle|1_a\rangle}{\sqrt{|r_0|^2 + |r_1|^2}}. \tag{2}$$

$|0_a\rangle$ and $|1_a\rangle$ represent the coupling and non-coupling atomic states, respectively, and $|\Psi_{\text{ph}}\rangle$ represents the photon polarization state, which experiences a rotation $\hat{R}$ when the atom is in state $|1_a\rangle$. $r_0$ and $r_1$ describe the field reflection coefficients as used in Methods section 'Atom–cavity interaction and conditional reflection'. Moreover, we assume no qubit rotation in case of a coupling atom. When the atomic part is not observed, the entanglement translates into decoherence for the photonic qubit as shown by the shrinking sphere in Extended Data Fig. 3c. This situation holds true only for polarization states that experience a rotation due to the cavity birefringence. Because photonic polarizations $A$ and $D$ do not rotate, for these polarizations the state in equation (2) becomes separable and therefore follows the ideal situation.

As mentioned previously, the polarization rotation leads to a qubit infidelity that could be overcome by means of retardation waveplates. However, in addition one would also expect a polarization dependency on the imprinted atomic phase shift at qubit reflection which would affect the nondestructive detection efficiency. We have tried to observe this effect by comparing the detector performance for different incoming polarization states (Fig. 3), but do not find substantial differences in the NPQD performance. We attribute this observation to the fact that the birefringence effect is relatively small compared to other imperfections, for example, the conditional reflection.

## Theoretical models for NPQD applications

In Fig. 4 we show simulations of specific situations in which our nondestructive detector is advantageous. The details of these simulations are explained in this section. As the operating wavelength of our NPQD is 780 nm, we consider photonic qubits of this wavelength with a corresponding fibre attenuation of $\alpha_{\text{att}} = 4$ dB km$^{-1}$.

**Situation 1.** Situation 1 (Fig. 4a) describes long-distance entanglement between an atom–photon entanglement source and a heralded quantum memory. For simplicity we consider the ideal situation, in which the time required to entangle the two systems is given by the communication time (for example, the time to distribute the entangled photonic qubit plus the time to communicate back if the heralded storage succeeded). In such a situation the mean entanglement time is given by $T_{\text{ent}} = 2L/(cp_{\text{ent}})$, where $L$ is the distance between sender and receiver, $c$ is the speed of light in an optical fibre and $p_{\text{ent}} = \eta_{\text{AP}}10^{-\alpha_{\text{att}}L/10}\eta_{\text{H}}$ is the heralded entanglement distribution probability. This probability depends on the atom–photon entanglement source efficiency $\eta_{\text{AP}}$, the attenuation coefficient of the transmission channel $\alpha$, and the heralding efficiency of the heralded quantum memory $\eta_{\text{H}}$.

When an NPQD is inserted along the transmission channel at a distance $l < L$ from the sender, the mean entanglement time reads $T_{\text{ent}}^{\text{NPQD}} = 2\langle l\rangle/(cp_{\text{ent}}^{\text{NPQD}})$. Compared with $T_{\text{ent}}$, the average communication distance $\langle l\rangle$ replaces $L$ and is given by $\langle l\rangle = P(0_a)L + [1 - P(0_a)](l + t_{\text{NPQD}}c/2)$. It has two terms: First, the NPQD may provide a nondestructive detection event that goes along with the full distance $L$. Second, the NPQD does not detect any qubit photon, leading to a shortening of the effective distance to $l$. In this case, one has to consider the NPQD herald signal readout time $t_{\text{NPQD}}$, which delays the subsequent classical communication. Both terms include the probability for a nondestructive detection event, which is

$$P(0_a) = \eta_{\text{AP}}10^{-\alpha_{\text{att}}l/10}P(0_a|1_{\text{iq}})(1 - p_{\text{DC}}) + p_{\text{DC}}. \tag{3}$$

It takes into account the nondestructive detection probability upon an incoming qubit $P(0_a|1_{\text{iq}})$ and the NPQD dark counts, $p_{\text{DC}}$. In addition to the replacement of $L$ by $\langle l\rangle$ in $T_{\text{ent}}^{\text{NPQD}}$, one needs to provide the entanglement probability at the presence of an NPQD, $p_{\text{ent}}^{\text{NPQD}}$, which replaces $p_{\text{ent}}$. It is given by $p_{\text{ent}}^{\text{NPQD}} = p_{\text{ent}}P(0_a|1_{\text{iq}})P(1_{\text{oq}}|0_a)$ that additionally considers the NPQD inefficiency. These expressions at hand, we calculate the entanglement time $T_{\text{ent}}^{\text{NPQD}}$ and compare it with the situation that does not include a nondestructive detector $T_{\text{ent}}$, in order to obtain the entanglement speed-up. For both simulations we consider $\eta_{\text{AP}} = 0.5$ and $\eta_{\text{H}} = 0.11$, which are realistic parameters that have been obtained in previous works[26,38].

**Situation 2.** Situation 2 (Fig. 4b) describes qubit amplification for different probabilities of input photon numbers. The qubit amplification $A_q$ is defined as the ratio of the probability of having a photonic qubit at the output of the NPQD conditioned on its nondestructive detection $P(1_{\text{oq}}|0_a)$ to the probability of having an input qubit $P_{\text{iq}}$. These quantities are related to those shown in Fig. 3, from which the qubit amplification is inferred as $A_q = \bar{n}_{\text{oq}}(0_a)/|\alpha|^2$ for small $|\alpha|^2$. The data points shown in Fig. 4b are calculated according to this expression. To simulate the qubit amplification with our NPQD parameters (solid line) and without NPQD dark counts (dashed line), we use the theoretical model described in Supplementary Information.

**Situation 3.** In situation 3 (Fig. 4c), remote photonic qubits are measured with noisy detectors: an NPQD gates a qubit measurement in order to improve its signal-to-noise ratio, SNR = $p_s/p_n$. This gating reduces the noise detection probability to $p_n^{\text{NPQD}} = P(0_a)p_n$ but also reduces the signal, owing to the parasitic losses induced by the nondestructive detector, $p_s^{\text{NPQD}} = p_sP(0_a|1_{\text{iq}})P(1_{\text{oq}}|0_a)$. Additionally, we relate the nondestructive detection probability with our NPQD parameters, $P(0_a) = P(0_a|1_{\text{iq}})p_s(1 - p_{\text{DC}}) + p_{\text{DC}}$, and consider that the input signal probability depends only on the transmission losses of the communication channel, $p_s = 10^{-\alpha_{\text{att}}L/10}$. This enables us to calculate the SNR gain as a

function of the communication distance $L$, considering an NPQD with our parameters (solid line) and a perfect NPQD (dashed line).

## Data availability

The datasets generated during and/or analysed during the current study are available at https://doi.org/10.5281/zenodo.4381767.

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

**Acknowledgements** We thank M. Brekenfeld and J. D. Christesen for contributions during an early stage of this work. This work was supported by Bundesministerium für Bildung und Forschung via Verbund Q.Link.X (grant number 16KIS0870), Deutsche Forschungsgemeinschaft under Germany's Excellence Strategy (EXC-2111, 390814868) and the European Union's Horizon 2020 Research and Innovation programme via the project Quantum Internet Alliance (GA number 820445). P.F. acknowledges support by the Cellex-ICFO-MPQ postdoctoral fellowship programme.

**Author contributions** All authors contributed to the experiment, analysis of the results and writing of the manuscript.

**Funding** Open access funding provided by Max Planck Society.

**Competing interests** The authors declare no competing interests.

**Additional information**
**Correspondence and requests for materials** should be addressed to D.N.

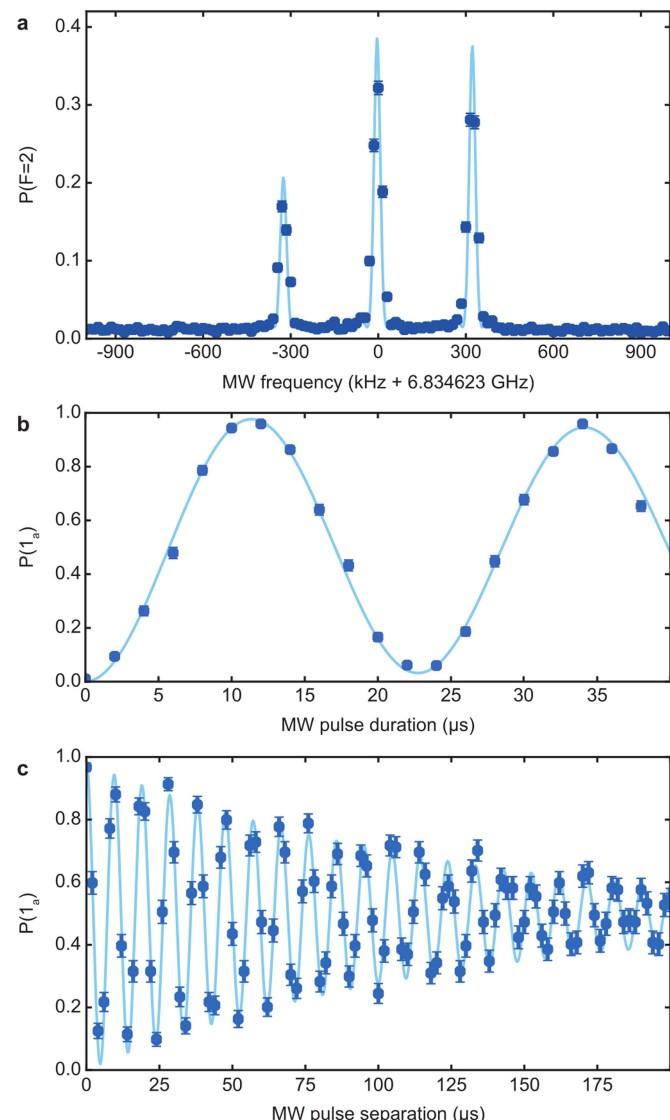

**Extended Data Fig. 1 | Coherent microwave driving. a**, Microwave spectroscopy of an atom prepared in Zeeman state $|F=1, m_F=0\rangle$. The light-blue curve is a fit consisting of three $\text{sinc}^2$ functions. Fit parameters are centre and Rabi frequency of each transition whereas the pulse duration is fixed to 25 µs. The peak at 0 kHz corresponds to the $|0_a\rangle \leftrightarrow |1_a\rangle$ transition. **b**, Microwave-driven Rabi oscillation on the $|0_a\rangle \leftrightarrow |1_a\rangle$ transition for an atom prepared in state $|0_a\rangle$. **c**, An atom prepared in state $|0_a\rangle$ is subsequently driven into a superposition of $|0_a\rangle$ and $|1_a\rangle$ by a microwave $\pi/2$ pulse. After a variable time a second microwave $\pi/2$ pulse is applied. During the waiting time the internal microwave clock is shifted by 100 kHz. Final state detection measures the population in $|1_a\rangle$. All error bars represent the $1\sigma$ confidence interval, some error bars are smaller than the symbol size.

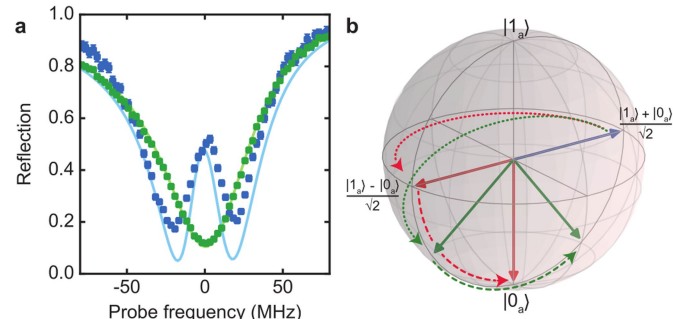

**Extended Data Fig. 2 | Atom–cavity reflection spectra and the atomic state during the NPQD scheme. a**, Cavity reflection spectra of the qubit cavity with no atom (green points) and with a cavity mode that is strongly coupled to an atom prepared in the state $|F=1, m_F=0\rangle$ (blue points). The solid lines represent fit functions and the error bars show the standard deviation. **b**, Bloch sphere with the atomic state (represented by vectors) at different stages of the detector scheme. The blue vector is the initial atomic state after the first $\pi/2$ microwave pulse. After photon reflection, the initial state turns into the red or green state (dotted arrow is a guide to the eye). Green and red states show the situation with and without conditional reflection, respectively. A subsequent microwave $\pi/2$ pulse rotates the atom to the final state (dashed arrow follows the rotation). State detection (not shown) then projects the atomic state onto $|0_a\rangle$ or $|1_a\rangle$.

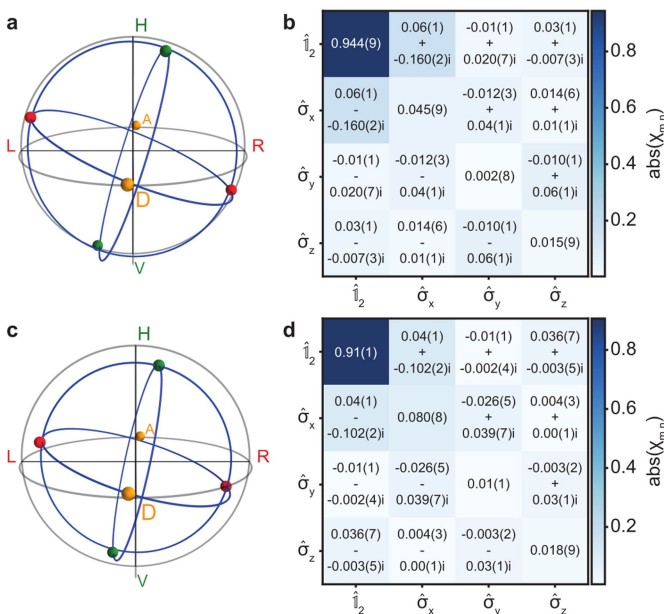

**Extended Data Fig. 3 | Polarization state and quantum process tomography. a**, **c**, Poincaré sphere showing the underlying quantum process of our NPQD. The coloured spheres are the result of polarization state tomography for reflected photonic pulses for the correspondingly coloured and labelled input polarizations. **b**, **d**, NPQD underlying quantum process matrix $\chi$ which is reconstructed via a maximum-likelihood fit. $\hat{\mathbb{1}}_2$ denotes the $2 \times 2$ identity matrix and $\hat{\sigma}_x$, $\hat{\sigma}_y$ and $\hat{\sigma}_z$ denote the Pauli matrices. In **a** and **b**, the photon counts at the tomography setup are conditioned on the nondestructive qubit detection. In **c** and **d**, they are not conditioned on the nondestructive detection. The rotation around the $A/D$ axis is described by an operator built by $\sigma_x$. The uncertainty of $\chi_{m,n}$ is assessed with a Monte Carlo method and represents the standard error.

**Extended Data Table 1 | Photonic polarization qubit fidelities**

| Polarization state | $\mathcal{F}_{\text{cond.}}$[*] | $\mathcal{F}_{\text{uncond.}}$[†] |
|:---:|:---:|:---:|
| $\lvert H\rangle$ | $0.94 \pm 0.01$ | $0.92 \pm 0.01$ |
| $\lvert V\rangle$ | $0.969 \pm 0.008$ | $0.92 \pm 0.01$ |
| $\lvert A\rangle$ | $0.985 \pm 0.004$ | $0.985 \pm 0.004$ |
| $\lvert D\rangle$ | $0.986 \pm 0.004$ | $0.978 \pm 0.004$ |
| $\lvert R\rangle$ | $0.944 \pm 0.009$ | $0.90 \pm 0.01$ |
| $\lvert L\rangle$ | $0.943 \pm 0.009$ | $0.910 \pm 0.009$ |
| **Fidelity** | | |
| $\overline{\mathcal{F}}_{\text{all}}$[‡] | $0.962 \pm 0.003$ | $0.936 \pm 0.003$ |
| $\overline{\mathcal{F}}_{\circlearrowleft}$[§] | $0.980 \pm 0.003$ | $0.944 \pm 0.003$ |
| $\mathcal{F}_{\chi}$[¶] | $0.944 \pm 0.009$ | $0.91 \pm 0.01$ |
| $\overline{\mathcal{F}}_{\text{s},\chi}$[**] | $0.963 \pm 0.006$ | $0.938 \pm 0.007$ |

The polarization state fidelities, $\overline{\mathcal{F}}_{\text{all}}$ and $\overline{\mathcal{F}}_{\circlearrowleft}$ are assigned with a 1σ confidence level accounting for statistical uncertainties due to the finite number of detected photons. The uncertainty of $\mathcal{F}_{\chi}$ is assessed with a Monte Carlo method and represents the standard error.
*SPD counts are conditioned on the nondestructive qubit detection.
†SPD counts are unconditionally evaluated.
‡Mean value of all six input polarization states.
§An inverse unitary rotation around the $A/D$ axis is applied to the measured output states before calculating the mean fidelity.
¶Quantum process fidelity.
**Mean state fidelity inferred from the quantum process fidelity via $\overline{\mathcal{F}}_{\text{s},\chi} = (2\mathcal{F}_{\chi} + 1)/3$.