## [Peer Review File · Nature]

Peer Review File**Manuscript Title:** Nondestructive detection of photonic qubits**Editorial Notes:****Redactions – Mention of other journals**

This document only contains reviewer comments, rebuttal and decision letters for versions considered at *Nature*. Mentions of the other journal have been redacted.

Reviewer Comments & Author Rebuttals**Reviewer Reports on the Initial Version:**

Referee #1 (Remarks to the Author):

In their manuscript, Niemietz et al. succeeded to trap a single atom in two crossed fibre-based optical resonators, the first one being used for atom-photon coupling, the second one facilitating the atom state detection. This experimental tour de force allows them to report on a non-destructive photonic qubit detector, a particularly interesting device in quantum information sciences enabling a large number of experiments for fundamental and applied physics. They provide a detailed characterisation of their detector in term of non-destructive detection efficiency, photon survival probability and fidelity. Their paper is well structured, nicely written, the motivations are clear, a fair comparison to previous works is done, many details are given in the method sections and in supplementary materials and the potential of their approach is largely discussed. Their manuscript deserves the attention of a broad audience and I would recommend publication in the prestigious journal *Nature* provided that the authors accept to answer the following comments.

1-Their detector has been tested and characterised with attenuated coherent states of light while some readers might expect an experimental demonstration with true single photons. Can the authors comment on this point?

2-To emphasise on the relevance of the first question, cloning of coherent states is possible provided they are large enough (in term of photon number) and hence, a reader might ask if the non-destructive photonic detection reported in the manuscript could be reproduced with a classical strategy (cloning + detection of the clones in different basis for example). Can the authors argue that their non-destructive photonic qubit detector operates in the quantum regime?

3-It takes about 20 microseconds to detect the state of the atom and to conclude on the presence/absence of a photon without disturbing its quantum state. This would correspond to several kilometres of fibres if we want to operate on the photon state only when we are sure it is indeed there (as in a Bell test). Given the attenuation of a fibre at 780 nm, is the heralding detector reported in the manuscript realistically useful for a Bell test and corresponding applications? Note that 43% is the minimum efficiency for atom-photon Bell test and this number increases with the error (non-unit fidelity).

4-I do appreciate the theoretical model in the methods section E aiming to show the usefulness of non-demolition photonic qubit measurements for generating sender-receiver entanglement. The

authors focus on the case with a single non-demolition photonic qubit measurement between the sender and receiver. What can they say about multiple uses of non-demolition photonic qubit measurements? How much does the communication time reduced in this case?

5-The fact that a non-demolition measurement increases the signal to noise implies that it could be useful to extend the distance where QKD can be performed (which is essentially limited by dark counts as far as I know). A comment could be added along this line.

6-The photon reflection when the atom sits in state $|0\rangle$ depends on the cooperativity C and I am a bit surprised that a reflection of 50% can be obtained with $C \leq 2$. My confusion likely comes from the definition of C . Can the authors remind me the correct formula, for example in the method section C?

7-I do appreciate very much the effort that the authors made to compare their results to the one reported in Ref. 18. I felt that the paragraph in the main text, however, could be improved. For example, taking about « the » newly added state-detection cavity is not a very clear way to mention a novel ingredient. Can the author improve the writing of this paragraph?

Referee #2 (Remarks to the Author):

The paper demonstrates experimentally the quantum non-demolition detection of a polarization encoded qubit using a sophisticated setup involving single atoms trapped at the crossing point of two very compact fiber cavities. Photons impinging on the "qubit cavity" are reflected with a probability of 35% leaving their imprint on one atom trapped at cavity center. The final atomic state heralding the presence of a reflected photon is readout with a very high fidelity (98%) using a second optical cavity ("state detection cavity"). The performance of the measurement is characterized by the strong correlation between state detection cavity signal and the actual detection of a reflected photon using a standard, destructive single photon detector.

Up to this point, the presented work is very similar in its principle to the one presented in ref 18, where comparable quantum non demolition detection performance was demonstrated. The main technical difference is the use of compact fiber cavity technology instead of using macroscopic mirrors. This provides a significant advantage wrt the case of ref 18 as already demonstrated in ref 25 , where the same setup was used in order to demonstrate the operation of an heralded quantum memory.

However, there is a major difference wrt to ref 18 in the fact that the authors now demonstrate the operation of QND measurement while preserving the quantum information of one qubit eventually encoded in the polarization of the reflected photon (Non-destructive Photonic Qbit Detector, NPQD). Using polarization state tomography, the authors demonstrate preservation of qubit encoded information with an impressive fidelity larger than 96%. The paper makes this result stronger by finally discussing four quantum information protocols, which may already benefit from the actual performance of the demonstrated NPQD.

In my opinion, the presented work is an extremely impressive experimental tour de force. With respect to previous achievements of similar setup, it demonstrates an essential new feature of high importance for quantum communication. It is also presented extremely clearly and very accessible for all the community of scientists interested in quantum information. As a result I strongly recommend publication of the paper in Nature.

Minor comment, which may be addressed by the authors (except point 2, which have to be addressed):

1- Page 7: the first sentence introducing the second example illustrating usefulness of NQPD is not very clear. The authors may state more directly that it is more efficient to perform complex operations on an input photonic qubit only if this qubit is present after transmission in a lossy channel. I find the mention of "long-distance entanglement" confusing.

2- Figure 1: The presence of NPBS with reflectivity 0,5% at the input of the NPQD may look like a severe limitation of the practical use of this detection method with precious single photons instead of weak coherent pulses. The authors should discuss in the main text how this difficulty can be overcome.

3- Figure 4: it would be clearer if the color code of curves was mentioned in the caption

4- Page 21: the discussion of entanglement time in term of distance is difficult to follow.

Supplementary information

5- Bottom of page 1: the definition of notation for the state in a given mode makes a confusion between "mode" and "state" in the sentence "After interaction, the photonic part might populate...."

6- Page 2, line 20: similar remark as above: I suggest to replace "the loss mode" by "the state of the loss mode".

7- Equation 2: I do not expect that the reflected states $r0_a$ and $r1_A$ are coherent states in general. I guess that it is only the case for weak pulses with negligible probability to have more than one photon. The authors may mention this point.

Referee #3 (Remarks to the Author):

The manuscript by D. Niemitz and coworkers reports on a quantum non-demolition measurement that allows heralding the presence of a photonic qubit without destroying its unknown quantum state. Such a measurement is important in quantum communications as it helps overcoming the effect of loss during qubit transmission. The measurement is well done and the results are convincing. However, I don't think that the advance compared to Kalb et al. in Phys. Rev. Lett. 114, 220501 (2015), in which the same group already reported heralded transfer of a photonic qubit state onto a single atom, as well as the reverse process (qubit transfer from an atom onto a single photon), is sufficient to warrant publication in Nature. Interestingly, this paper is not even mentioned in the manuscript. In short, I suggest re-submission to a more technical journal such as [REDACTED] or [REDACTED].

Author Rebuttals to Initial Comments:

Referee #1 :

In their manuscript, Niemietz et al. succeeded to trap a single atom in two crossed fibre-based optical resonators, the first one being used for atom-photon coupling, the second one facilitating the atom state detection. This experimental tour de force allows them to report on a non-destructive photonic qubit detector, a particularly interesting device in quantum information sciences enabling a large number of experiments for fundamental and applied physics. They provide a detailed characterisation of their detector in term of non-destructive detection efficiency, photon survival probability and fidelity. Their paper is well structured, nicely written, the motivations are clear, a fair comparison to previous works is done, many details are given in the method sections and in supplementary materials and the potential of their approach is largely discussed. Their manuscript deserves the attention of a broad audience and I would recommend publication in the prestigious journal Nature provided that the authors accept to answer the following comments.

We thank the referee for the helpful comments and for the recommendation to publish our work in Nature. We are happy reading that our NPQD “is a particularly interesting device in quantum information sciences” and that our manuscript “deserves the attention of a broad audience”.

1) Their detector has been tested and characterised with attenuated coherent states of light while some readers might expect an experimental demonstration with true single photons. Can the authors comment on this point ?

Response : We agree with the comment of the referee. Testing our NPQD with true single photons would have had the additional advantage that the measurement count rate would have been significantly higher since a weak coherent pulse provides a non-vacuum contribution with only a small probability, much less than 1. Compared to attenuated laser pulses, a single photon source would have provided a signal in almost every experimental trial. However, at the time of data acquisition we did not have a single photon source available. Therefore, we used weak coherent pulses at the single photon level. We do not expect fundamental differences as long as the mean photon number of the weak coherent states is so small that higher photon number components are small compared to the single photon component.

We emphasize that we have thoroughly characterized and discussed the consequences the usage of weak coherent pulses has on the performance of the quantum detector (see Fig. 3 and line 155 - 179 in the main text), including the range of $|\alpha|^2$ where higher photon number contributions are not negligible.

We have also shown that the quantum detector does not detect the incoming overall weak coherent pulse but only the single photon component, given $|\alpha|^2 \ll 1$. This comes from a projective measurement that distills a single photon out of a coherent pulse. We have verified this with a measurement of the second order correlation function of the output field, see inset Fig. 3b.

2) To emphasise on the relevance of the first question, cloning of coherent states is possible provided they are large enough (in term of photon number) and hence, a reader might ask if the non-destructive photonic detection reported in the manuscript could be reproduced with a classical strategy (cloning + detection of the clones in different basis for example). Can the authors argue that their non-destructive photonic qubit detector operates in the quantum regime ?

Response : It is not possible to construct a device that creates an exact copy of an arbitrary quantum state [Wootters *et al.*, Nature 299, 802-803 (1982)]. However, one could ask the question what the upper bound for the fidelity of an universal quantum cloning machine is. Scarani *et al.* [Rev. Mod. Phys. 77, 1225 (2005)] provide the following relation for the maximally achievable fidelity F with N input and M output qubits with $N \leq M$:

$$F_{N \rightarrow M} = \frac{MN + M + N}{M(N + 2)}. \quad (1)$$

A rather simple example for a cloning machine is to measure the incoming qubits, store the result classically and generate new qubits based on the obtained measurement result. In the described scenario the number of outgoing qubits can be infinitely high ($M \rightarrow \infty$) due to the classical knowledge of the bits. Equation 1 is therefor $F_{M \rightarrow \infty} = (N + 1)/(N + 2)$ and yields an upper limit of 2/3 in case of a single incoming qubit. Specht *et al.* [Nature 473, 190-193 (2011), SI] consider this as a classical limit for quantum memories and extend it to the use of weak coherent states which increase the limit due to higher photon number contributions. For a weak coherent pulse as we have used it with mean photon number $|\alpha|^2 = 0.2$, the upper bound is shifted from 66.6 % to 67.5 %.

Another option is to use a $N = 1 \rightarrow M = 2$ cloning machine [Simon *et al.* Phys. Rev. Lett. 84, 2993 (2000)] where one clone serves as a herald photon. The maximum fidelity in this case is $5/6 \approx 83\%$. For weak coherent pulses, one could think of a hypothetical scenario where the single photon component is cloned (with maximum fidelity of 5/6) but higher photon components are split, taking one of the photons as a herald. The subtracted photon serves as a herald and the other part carries the full qubit information which goes along with a fidelity of one. In the end, the fidelity limit for weak coherent pulses would again increase and would yield the limit of 84.9 % for a mean photon number $|\alpha|^2 = 0.2$.

Depending on the cloning machine the fidelity limit is different. However, all scenarios discussed above come with fidelities that are significantly lower than what we have achieved in our work. Therefore, we conclude that our nondestructive detector operates in the quantum regime and cannot be replaced by concepts employing qubit cloning machines.

We have added a sentence in the main text, line 134 which summarizes the previous points.

3) It takes about 20 microseconds to detect the state of the atom and to conclude on the

presence/absence of a photon without disturbing its quantum state. This would correspond to several kilometres of fibres if we want to operate on the photon state only when we are sure it is indeed there (as in a Bell test). Given the attenuation of a fibre at 780 nm, is the heralding detector reported in the manuscript realistically useful for a Bell test and corresponding applications? Note that 43% is the minimum efficiency for atom-photon Bell test and this number increases with the error (non-unit fidelity).

Response : We agree with the referee that in some situations one has to operate on the photonic qubit only after the nondestructive detection signal is received. The read out time of the here presented quantum detector (given by the MW $\pi/2$ pulse and subsequent state detection) is $15 \mu\text{s}$ and therefore one option is to send the photonic qubit over a few kilometres of fiber during this time. This can indeed be a problem in a detection-loophole-free Bell test, as the fiber attenuates the qubit photon signal which, in the end, lowers the overall efficiency from 56 % to below 43 %.

However, our nondestructive detection time of $15 \mu\text{s}$ can be dramatically improved as we did not put much effort in minimizing it. Our group [Hacker *et al.* Nature 536, 193–196 (2016)] has successfully demonstrated that a $\pi/2$ pulse for atomic state manipulation can be realized within $1 \mu\text{s}$ with a Raman transfer. Moreover, that work demonstrates fluorescence state detection within $1.2 \mu\text{s}$ at a fidelity of 96 %. We are confident that in our setup we can realize state rotation and detection similarly fast without major experimental changes. In addition, the state detection duration can be further reduced by replacing the employed avalanche photodiode with a detection efficiency of $\sim 50\%$ by a superconducting nanowire single-photon detector which reaches an efficiency of $> 90\%$. Due to this, the state detection duration can be reduced by a factor $5/9$. Another aspect is the larger cooperativity C of the state detection transition with our state detection cavity. At current status, our Purcell factor $2C + 1$ is 35.2 % larger than the one which is given in the work of Hacker *et al.*. This yields a faster fluorescence photon emission out of the cavity. If we put effort in minimizing the state detection duration, the cooperativity aspect would certainly help us to fall below the duration stated in Hacker *et al.*. Moreover, we do not see a fundamental limitation to perform the atomic state rotations faster than $1 \mu\text{s}$. This would reduce the optical losses of a stored photon even more.

In addition to the strategy to shorten the NPQD read out time one could also follow the strategy to reduce the photonic losses that occur at the NPQD interaction. This would increase the outgoing mean photon number conditioned on a nondestructive detection event, $\bar{n}(0_a)$, which is the figure of merit that we compare to the minimum Bell test efficiency. One approach would be to increase the fibre-cavity mode matching which we estimate to be approximately 92 %. However, calculations about the mode matching that consider geometric parameters of the present in-/outcoupling cavity mirror surface, yield a maximum mode matching of $\approx 99\%$. The difference is attributed to a spatial misalignment of the cavity fibers which we can adjust via the already implemented piezo translation stages.

Another aspect is the increase of the minimum efficiency of the Bell test due to a non-unit qubit fidelity. We attribute the 2 % infidelity, as already mentioned in the main text,

to calibration errors of the qubit measurement setup and to fluctuations of the qubit cavity resonance frequency. Both points can be improved and are not subject to fundamental limitations.

In conclusion, we have discussed a group of possible improvements that do not require major changes of the current system. We are confident that, if we push above discussed parameters to a limit, we exceed the minimum efficiency required for a Bell test measurement, including the NPQD read out time.

4) I do appreciate the theoretical model in the methods section E aiming to show the usefulness of non-demolition photonic qubit measurements for generating sender-receiver entanglement. The authors focus on the case with a single non-demolition photonic qubit measurement between the sender and receiver. What can they say about multiple uses of non-demolition photonic qubit measurements? How much does the communication time reduced in this case?

Response : The referee is correct in expecting that multiple NPQDs along a transmission line can reduce further the communication time. However this is only true for a high qubit fidelity and nondestructive detection efficiency. For a better understanding, we extended the theoretical model given in Method E to include the usage of multiple NPQDs. In the related numerical calculation the sender-receiver distance was chosen to be 15 km, similar to the distances that are considered in the main text and Fig. 4a. For simplicity, we have chosen the locations of the quantum detectors such that they have equal distance to each other as well as to the sender and to the receiver. The simulation considers ideal NPQDs except for $P(0_a|1_{iq})$, i.e. the heralding efficiency given a photon at the input of the detector, which was chosen to be 1, 0.9 or 0.7. Fig. 1 shows that for ideal NPQDs ($P(0_a|1_{iq}) = 1$) the entanglement time continuously decreases and therefore the entanglement speed-up increases with a growing number of detectors. However, NPQDs with imperfect efficiencies ($P(0_a|1_{iq}) = 0.9$ or 0.7) lead to a decrease of the entanglement time only until a certain number of NPQDs (6 or 2, respectively). For larger numbers, the additional NPQD losses are too large to speed up the entanglement time.

From the above discussion one can anticipate that the efficiency of our NPQD, $P(0_a|1_{iq}) = (45 \pm 2) \%$, is not high enough to reduce the entanglement time in case of multiple NPQDs on a sender-receiver distance of 15 km. Moreover, further imperfections will have an additional negative effect and strengthen the hypothesis that our current NPQD is not yet suitable for the concatenation in a transmission line at a total distance on the order of tens of kilometres. We do expect, however, an improvement for much larger distances and tailored NPQD locations, although the absolute entanglement time becomes in this situation too long to be practical.

FIGURE 1 – **Multiple NPQDs used for long distance entanglement.** As a figure of merit, the entanglement speed-up and the entanglement time is provided for different numbers of employed NPQDs. The nondestructive detectors are distributed with equal distance to each other and to the sender and to the receiver. The sender-receiver distance is chosen to be 15 km. The NPQDs are considered to be ideal except for the efficiency $P(0_a|1_{iq})$, which we assume to be 1, 0.9 or 0.7 (see data labels). The blue (green) coloured lines refer to the entanglement speed-up (entanglement time).

5) *The fact that a non-demolition measurement increases the signal to noise implies that it could be useful to extend the distance where QKD can be performed (which is essentially limited by dark counts as far as I know). A comment could be added along this line.*

Response : We appreciate the suggestion and added this point to the main text, line 203.

6) *The photon reflection when the atom sits in state $|0\rangle$ depends on the cooperativity C and I am a bit surprised that a reflection of 50% can be obtained with $C \leq 2$. My confusion likely comes from the definition of C . Can the authors remind me the correct formula, for example in the method section C?*

Response : The cooperativity is defined by $C = g^2/2\kappa\gamma$ as given in Methods section C, line 355. The intensity reflection coefficient at zero detuning can be expressed as a function of the cooperativity as $|r(\Delta = 0)|^2 = \left|1 - \mu_{fc}^2 \frac{2\kappa_1}{\kappa} \frac{1}{2C+1}\right|^2$. $\mu_{fc} = 0.92e^{-i0.03}$ describes the mode matching between the fiber and the cavity mode. κ is the overall cavity field decay rate and κ_1 describes the cavity field decay rate via the outcoupling mirror. The ratio κ_1/κ provides the outcoupling ratio of 79%, a typical value for single-sided cavities. With the cooperativity $C = 1.67 \pm 0.09$ and the equation presented here one can reach a value very close to 50%. We have added a sentence in Method C, line 357 which provides the above-written equation for the intensity reflection coefficient as a function of the cooperativity.

7) *I do appreciate very much the effort that the authors made to compare their results to the one reported in Ref. 18. I felt that the paragraph in the main text, however, could be improved. For example, taking about « the » newly added state-detection cavity is not a very clear way to mention a novel ingredient. Can the author improve the writing of this paragraph?*

Response : We agree with the referee that the paragraph, especially the description of the second key ingredient, can be improved. A second crossed cavity with small mode volume is mainly enabled by miniaturized fibre resonators. We have added this information to the paragraph (line 97).

Referee #2 :

The paper demonstrates experimentally the quantum non-demolition detection of a polarization encoded qubit using a sophisticated setup involving single atoms trapped at the crossing point of two very compact fiber cavities. Photons impinging on the “qubit cavity” are reflected with a probability of 35% leaving their imprint on one atom trapped at cavity center. The final atomic state heralding the presence of a reflected photon is readout with a very high fidelity (98%) using a second optical cavity (“state detection cavity”). The performance of the measurement is characterized by the strong correlation between state detection cavity signal and the actual detection of a reflected photon using a standard, destructive single photon detector. Up to this point, the presented work is very similar in its principle to the one presented in ref 18, where comparable quantum non demolition detection performance was demonstrated. The main technical difference is the use of compact fiber cavity technology instead of using macroscopic mirrors. This provides a significant advantage wrt the case of ref 18 as already demonstrated in ref 25, where the same setup was used in order to demonstrate the operation of an heralded quantum memory.

However, there is a major difference wrt to ref 18 in the fact that the authors now demonstrate the operation of QND measurement while preserving the quantum information of one qubit eventually encoded in the polarization of the reflected photon (Non-destructive Photonic Qbit Detector, NPQD). Using polarization state tomography, the authors demonstrate preservation of qubit encoded information with an impressive fidelity larger than 96%. The paper makes this result stronger by finally discussing four quantum information protocols, which may already benefit from the actual performance of the demonstrated NPQD.

In my opinion, the presented work is an extremely impressive experimental tour de force. With respect to previous achievements of similar setup, it demonstrates an essential new feature of high importance for quantum communication. It is also presented extremely clearly and very accessible for all the community of scientists interested in quantum information. As a result I strongly recommend publication of the paper in Nature.

Minor comment, which may be addressed by the authors (except point 2, which have to be addressed) :

We thank the referee for the strong recommendation to publish our work in Nature. He/she describes our work as an “extremely impressive experimental tour de force”, finds that our manuscript is “presented extremely clearly and very accessible for all the community of scientists interested in quantum information”, and expresses the opinion that the use of compact fiber technology provides significant advantage with respect to macroscopic mirrors, building a major difference to Reiserer *et al.* [Science 342, 1349-1351 (2013)].

1) Page 7 : the first sentence introducing the second example illustrating usefulness of NPQD is not very clear. The authors may state more directly that it is more efficient to perform complex operations on an input photonic qubit only if this qubit is present after transmission in a lossy channel. I find the mention of “long-distance entanglement” confusing.

Response : We thank the referee for this comment. We have incorporated the advice in the

main text, line 190 and improved the description of the second example in order to avoid confusion.

2) Figure 1 : The presence of NPBS with reflectivity 0.5% at the input of the NPQD may look like a severe limitation of the practical use of this detection method with precious single photons instead of weak coherent pulses. The authors should discuss in the main text how this difficulty can be overcome.

Response : We agree with the referee. The highly transmitting NPBS was suitable for the characterization of the NPQD with weak coherent pulses. However, the nondestructive detection of true single photon qubits would greatly benefit from the replacement of the NPBS by a less lossy element, such as an optical circulator or a fast optical switch. This point is already considered in Fig. 1a. Additionally, we have added a sentence in the main text, line 104.

3) Figure 4 : it would be clearer if the color code of curves was mentioned in the caption

Response : We have used multiple colours in subfigure (a) only. Here, the colour is either assigned to the entanglement speed-up or to the NPQD entanglement time. The axis labels are correspondingly coloured to clarify the relation. To improve the clarity of the figure, we added a sentence to the caption.

4) Page 21 : the discussion of entanglement time in term of distance is difficult to follow.

Response : We thank the referee for the comment. We reworded the according paragraph in order to improve the readability.

Supplementary information

5) Bottom of page 1 : the definition of notation for the state in a given mode makes a confusion between “mode” and “state” in the sentence “After interaction, the photonic part might populate ...”

Response : We agree with the referee and have removed the ket-notation in the respective paragraph.

6) Page 2, line 20 : similar remark as above : I suggest to replace “the loss mode” by “the state of the loss mode”.

Response : We agree with the referee and added “the state of” in line 20. For the same reason we applied similar changes in lines 38 and 44.

7) Equation 2 : I do not expect that the reflected states r_{0_a} and r_{1_a} are coherent states in general. I guess that it is only the case for weak pulses with negligible probability to have more than one photon. The authors may mention this point.

Response : We agree with the referee on the validity for weak coherent pulses. We have given this condition in the Supplementary Information, second paragraph : “We consider weak coherent pulses in front of the qubit cavity $|\alpha\rangle$ that are interacting with the atom-cavity system.”. To emphasize the condition, we have added the word “weak” in front of “coherent field/pulse” in lines 17, 30 and in the second paragraph, third sentence.

Referee #3 :

*The manuscript by D. Niemietz and coworkers reports on a quantum non-demolition measurement that allows heralding the presence of a photonic qubit without destroying its unknown quantum state. Such a measurement is important in quantum communications as it helps overcoming the effect of loss during qubit transmission. The measurement is well done and the results are convincing. However, I don't think that the advance compared to Kalb *et al.* in *Phys. Rev. Lett.* 114, 220501 (2015), in which the same group already reported heralded transfer of a photonic qubit state onto a single atom, as well as the reverse process (qubit transfer from an atom onto a single photon), is sufficient to warrant publication in *Nature*. Interestingly, this paper is not even mentioned in the manuscript. In short, I suggest resubmission to a more technical journal such as [REDACTED] or [REDACTED].*

We appreciate that the referee finds that “the measurement is well done and the results are convincing”. The referee is, however, concerned about the advance of our work compared to the heralded photonic quantum memory scheme reported in Kalb *et al.*. As we argue in the following paragraphs, the advance is substantial and a comparison to the previous work is/was indirectly provided in our manuscript.

In our manuscript we did not directly compare our work to the photonic quantum memory reported in Kalb *et al.* because we already compare it to the one reported in Brekenfeld *et al.* (ref. 26). Ref. 26 reports a heralded photonic quantum memory that is more advanced and shows a significantly better performance than the one reported in Kalb *et al.*. The comparison of our work to a heralded quantum memory can be read in lines 64-70 of the main text, where the differences and the novelty of our NPQD are highlighted. In order to clarify this point in more detail, we extend the comparison in the following paragraph.

While it is true that the reported heralded photonic quantum memories can be used to herald photonic qubits, both schemes have major differences and severe limitations compared to our NPQD. As we mention in the main text, they provide only a single photon as a herald signal, which is very sensitive to optical losses. Consequently, the herald efficiency is significantly worse compared to our scheme. Moreover a heralded photonic quantum memory is fundamentally different from a NPQD in the sense that it is based on the absorption and later re-emission of a qubit photon. This process does not maintain the photonic pulse shape which is of high importance for time bin qubits that we anticipate to nondestructively detect with our approach as well. In addition, the qubit/herald photon in Kalb *et al.* may have not interacted with the atom-cavity system due to an imperfect cavity mode matching. This renders the herald signal not only inefficient but also unreliable.

In the previous version of the manuscript we decided to only cite Brekenfeld *et al.*, line 65, since the demonstrated experiment is conceptually and quantitatively a better heralded quantum memory, and the number of references for our manuscript is limited. However, as we now realized that this point may confuse readers that are more familiar with Kalb *et al.* than with the more recent Brekenfeld *et al.*, we now include both citations in the paragraph (line 65). The arguments given in the paragraph are suitable to the comparison of both articles, and therefore no major changes in the paragraph are required.

Reviewer Reports on the First Revision:

Reviewer #1 provided confidential remarks to the editor, supporting publication of your manuscript.

Referee #2 (Remarks to the Author):

I think that the authors have properly taken into account my comments as well as the one formulated by referee #1.

Concerning the relation of the present work with respect to Kalb et al. in Phys. Rev. Lett. 114, 220501 (2015), (as raised by referee #3), I do not think that the present work is a minor improvement. As stated by the authors, a heralded memory is not equivalent to the presented QND measurement scheme. Stacking the error of writing and readout is by far less efficient and the effective emission of readout photon is not even heralded. In addition, the fact that the experiment is based on miniature, fiber cavities, is not anecdotic. I think that it would have been utterly difficult to achieve the present goal with macroscopic cavities as in the case of Kalb et al. The use of fiber cavity technology may be a breakthrough for implementing complex quantum communication protocols.

As a conclusion, I would like to say that I consider the presented work as a beautiful experimental achievement in itself. Independently of promises for future applications, I think that it defines a new "state of the art" for optical cavity QED and I think that it fully deserves publication in Nature.

Referee #3 (Remarks to the Author):

I am happy that (and how) the authors clarified the difference with respect to previous experiments by the same group and recommend now publication of their manuscript in Nature.